# Toggle-like current-induced Bloch point dynamics of 3D skyrmion strings in a room temperature nanowire

M. T. Birch [1✉], D. Cortés-Ortuño[2✉], K. Litzius [1], S. Wintz[1,3], F. Schulz[1], M. Weigand [3], A. Štefančič[4,5], D. A. Mayoh [4], G. Balakrishnan [4], P. D. Hatton [6] & G. Schütz[1]

Research into practical applications of magnetic skyrmions, nanoscale solitons with interesting topological and transport properties, has traditionally focused on two dimensional (2D) thin-film systems. However, the recent observation of novel three dimensional (3D) skyrmion-like structures, such as hopfions, skyrmion strings (SkS), skyrmion bundles, and skyrmion braids, motivates the investigation of new designs, aiming to exploit the third spatial dimension for more compact and higher performance spintronic devices in 3D or curvilinear geometries. A crucial requirement of such device schemes is the control of the 3D magnetic structures via charge or spin currents, which has yet to be experimentally observed. In this work, we utilise real-space imaging to investigate the dynamics of a 3D SkS within a nanowire of $Co_8Zn_9Mn_3$ at room temperature. Utilising single current pulses, we demonstrate current-induced nucleation of a single SkS, and a toggle-like positional switching of an individual Bloch point at the end of a SkS. The observations highlight the possibility to locally manipulate 3D topological spin textures, opening up a range of design concepts for future 3D spintronic devices.

[1] Max Planck Institute for Intelligent Systems, 70569 Stuttgart, Germany. [2] Department of Earth Sciences, Utrecht University, 3584 CB Utrecht, The Netherlands. [3] Helmholtz-Zentrum Berlin für Materialien und Energie GmbH, 12489 Berlin, Germany. [4] Department of Physics, University of Warwick, Coventry CV4 7AL, UK. [5] Electrochemistry Laboratory, Paul Scherrer Institut, CH-5232 Villigen, PSI, Switzerland. [6] Department of Physics, Durham University, Durham DH1 3LE, UK. ✉email: birch@is.mpg.de; d.i.cortes@uu.nl

In systems with broken inversion symmetry and the presence of strong spin-orbit coupling, the interplay of the Dzyaloshinskii-Moriya interaction (DMI) with the exchange interaction results in the formation of monochiral spin textures, including magnetic skyrmions[1–3]. In 2D thin-film systems, the symmetry-breaking proximity of heavy metal and ferromagnetic layers induces an interfacial DMI, resulting in the formation of Néel-type skyrmions[4,5], which are inherently limited in vertical size by the film thickness. On the other hand, in bulk chiral magnets, the broken inversion symmetry of the underlying crystal structure results in an isotropic bulk DMI[6], and the formation of extended skyrmion strings (SkS), which could conceptually extend through the thickness of a single crystal[7–10].

The extended vertical structure of a SkS was only recently identified in real-space in B20 chiral magnets[11–13], and subsequently via 3D tomographic imaging[14,15]. These SkS structures exhibit fascinating dynamical properties. Firstly, their formation and annihilation mechanisms are governed by topological defects known as Bloch points[7,8], which due to their singular nature, can be thought of as zero-dimensional domain walls. The motion of these emergent magnetic monopoles acts as a source or sink of topological charge, mediating the creation of 3D topological structures from trivial magnetic states[16,17]. Secondly, recent studies investigating the resonant and singular spinwave dynamics of SkSs have demonstrated the transmission of signals along the length of a SkS over length scales of 50 μm[18,19], indicating their potential use as nanoscale magnonic transmission lines[20].

Moreover, exotic 3D magnetic objects with higher-order topologies have recently been observed, such as the magnetic hopfion[21,22] or bound states composed of multiple SkSs in the form of skyrmion bundles[23], and skyrmion braids[24]. The potential to devise and assemble dynamic structures using these complex topological states in 3D or curvilinear geometries is a fascinating possibility[25], with the potential to realise improved spintronic device designs[26–29]. Current-induced motion and writing of individual skyrmions has been demonstrated for 2D-like skyrmions[30,31] (which we shall define as having vertical size less than their lateral size). However, such control has yet to be observed for 3D topological structures in real-space, despite previous transport measurements indicating this possibility[8,32]. Therefore, we identified a compelling need to explore the possibility of controlling and manipulating 3D SkS states in device-like geometries.

In this work, we utilise scanning transmission x-ray microscopy (STXM) to investigate the stability of SkS states in a nanowire of $Co_8Zn_9Mn_3$ at room temperature. We find that SkSs within the confined nanowire shape appear to be more stable than in plate-like lamellae samples. Furthermore, we demonstrate current-induced generation and local manipulation of an individual SkS, in the form of toggle-like switching of the SkS length. The results demonstrate the potential of future 3D skyrmion-based device schemes, and motivate further exploration to control the rich dynamics of 3D topological magnetic structures.

## Results

**Skyrmion strings in a nanowire.** Using a focused ion beam, we fabricated a nanowire with dimensions $5000 \times 700 \times 200$ nm from a single crystal of $Co_8Zn_9Mn_3$, a bulk chiral magnet with a measured Curie temperature, $T_C$, of ~360 K[33] (magnetometry characterisation and wire geometry shown in Supplementary Fig. S1). The sample was fixed to gold contacts patterned on a $Si_3N_4$ membrane, allowing current to be applied along its length (see methods). A schematic illustration and a scanning electron micrograph of the fabricated device is shown in Fig. 1a, b, respectively. A controllable quadrapole magnet allowed the field

to be applied either in the out-of-plane (OOP) or in-plane (IP) directions, respectively along the thickness (z-axis) and width (y-axis) of the wire, as indicated in Fig. 1a. We utilised an additional resistive wire attached to the sample holder to enable heating, and subsequent zero-field cooling (ZFC) or field-cooling (FC), of the nanowire.

Scanning transmission x-ray microscopy (STXM) was utilised to image the spin textures on the nanoscale. Magnetic contrast was achieved by exploiting the effects of x-ray magnetic circular dichroism (XMCD) at the resonant absorption Co $L_3$ edge at 779 eV, producing a signal proportional to $m_z$ (see methods, Supplementary Fig. S1). We identified four magnetic states within the nanowire: the helical state, formed after ZFC, with winding vector $k$ aligned along the x-axis, and real-space winding length 120 nm (Fig. 1c); the conical state, realised by applying an IP field, with winding length 120 nm and $k$ along the y-axis (Fig. 1d); the SkL state, produced after FC with an OOP field, showing its characteristic hexagonal structure with a skyrmion separation of 148 nm (Fig. 1e); and finally the SkS state, produced after FC with an IP field (Fig. 1f). The two SkSs can be identified by the light and dark vertical contrast lines, corresponding to the Bloch-type chirality, embedded within the horizontal contrast lines of the background conical state[11]. Their observation upon FC indicates that the SkSs were created within the equilibrium skyrmion phase close to $T_C$ during the cooling procedure, and quenched to low temperatures, forming a metastable state[34,35]. For each STXM image, a comparable micromagnetic simulation is shown. The plots show the $m_z$ component of the simulated system averaged through all thickness layers, demonstrating a strong agreement with the experimentally observed states (see methods). Visualisations of the SkL and SkS state simulations are shown in Fig. 1g, h respectively, highlighting the Bloch-type chiral structure of the skyrmions, and indicating the locations of the Bloch points at the termination points of the SkSs[8]. In reality, the size of the Bloch point is on the order of a few spins, and is therefore not resolvable with the spatial resolution of x-ray microscopy. However, the comparison of simulated and STXM images allows for the presence of a Bloch to be inferred within the experimental sample.

**Field-induced skyrmion string unwinding.** We explored the formation and stability of these chiral states within the confined nanowire by acquiring images as a function of increasing applied field for both OOP and IP orientations, starting at − 250 mT. The results are summarised in the single-temperature phase diagrams in Fig. 2a, b. The images utilised to create these diagrams are shown in Supplementary Fig. S2-4. For the OOP configuration, the helical state at 0 mT is transformed into a few isolated skyrmions at higher fields. This is in contrast to the SkL state shown in Fig. 1e, where skyrmions fill the entire nanowire because they were quenched to room temperature from the high temperature SkL phase during the FC process under an out-of-plane applied field. For an IP field, the helical state rotates to form the conical state, before reaching a uniformly magnetised (UM), saturated state above 100 mT. Fig. 2c displays the states realised after FC at different applied IP fields. When cooling within the range of 40 to 80 mT, we achieved SkS states embedded within the conical background. In one case, the same SkS state was observed to remain intact after 12 hours, indicating the formation of long-lived metastable SkSs[34,35].

Having achieved SkS states within the wire, we explored their stability against both increasing and decreasing applied magnetic field. Fig. 3a displays a STXM image of two adjacent SkSs within the conical background. When applying an increasing magnetic field, the conical state contrast disappeared, and the SkSs were shortened, demonstrating the topological unwinding process as the Bloch

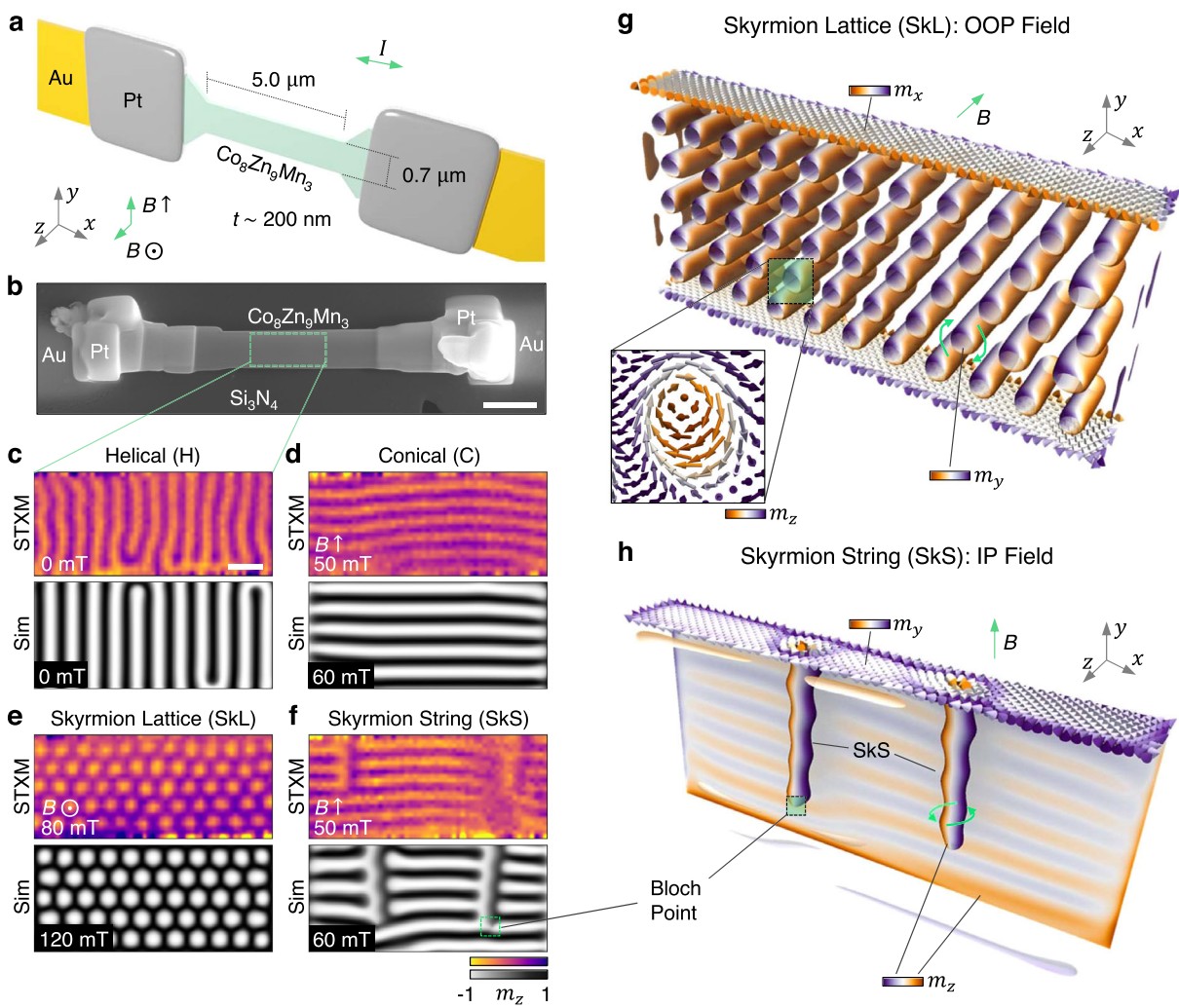

**Fig. 1 Co$_8$Zn$_9$Mn$_3$ nanowire and observed chiral spin textures. a** Schematic illustration of the device construction, showing the Co$_8$Zn$_9$Mn$_3$ nanowire with thickness $t$ of 200 nm, on the Si$_3$N$_4$ membrane, fixed by platinum (Pt) deposition welds to the patterned gold (Au) contacts. The orientations of the out-of-plane (OOP, $B \odot$) and in-plane (IP, $B\uparrow$) applied field configurations are indicated. The current flow direction ($I$) is indicated. **b** Scanning electron microscopy image of the final device. The scale bar is 1 μm. **c**–**f** Scanning transmission x-ray microscopy (STXM) and simulated (Sim) images of the helical (H), conical (C), skyrmion lattice (SkL) and skyrmion string (SkS) states, as observed in the nanowire. The colourmaps indicate the out-of-plane magnetisation component, $m_z$, and the scale bar is 250 nm. The small region with a lack of conical contrast in **f** is due to a rearrangement of the conical stripes between the acquisition of each x-ray polarisation image. **g**, **h** Three-dimensional visualisations of the simulated SkL and SkS states, realised with OOP and IP applied fields, respectively. The colourmap specifies the local orientation of the magnetisation ($m_{x,y,z}$), as indicated by the labels. The inset shows the local spin structure of the skyrmions in **g**.

point travelled along their length, as shown in Fig. 3b, c. Finally, at 110 mT, the SkS were reduced to a length comparable to the winding length of the material, indicating the formation of two chiral bobbers[36]. Therefore, the image in Fig. 3d presents a side-on perspective of these topological surface states. Fig. 3e–h display simulated images of comparable magnetic states, confirming the identity of the observed SkS and chiral bobber states.

We performed similar measurements for decreasing magnetic field, starting with a state consisting of two separated SkSs, as shown in Fig. 3i-l. For decreasing field, the SkSs annihilate into the conical/helical background by the formation of a dislocation, as shown in the case of the left SkS in Fig. 3k. Finally, the SkSs are totally annihilated into a mixed conical and helical state, shown in Fig. 3l. Once again, the simulated images in Fig. 3m-p demonstrate a comparable behaviour in the micromagnetic system (experimental data set and 3D visualisations of the simulations presented in Supplementary Fig. S5-7). Both high-field unwinding and low-field dislocation annihilation mechanisms are mediated by the

formation of Bloch point structures[7,8,17]. The presence of these diverging magnetisation points is shown in the 3D visualisations of the simulations presented in Fig. 3q, r.

In addition to the strong qualitative agreement of the simulated states with the experimental images, another key identifying property of the SkS state is their spacing $d$ as a function of the applied magnetic field. In particular, the helical and conical domains exhibit a consistent spacing of 120 nm, whereas the spacing of the SkS should be greater than 120 nm, and increase as a function of the applied field[11,37]. The spacing of the observed conical and SkS states was determined by acquiring line profiles of the experimental images, and plotted in Fig. 3s. The larger, field dependent spacing of the vertical contrast structures confirms their identity as SkS states. Similar behaviour of the spacing of each state was reproduced in the simulated system (examples shown in Supplementary Fig. S8).

Due to shape anisotropy and confinement effects, the dimensions of the host system play a crucial role in the stability

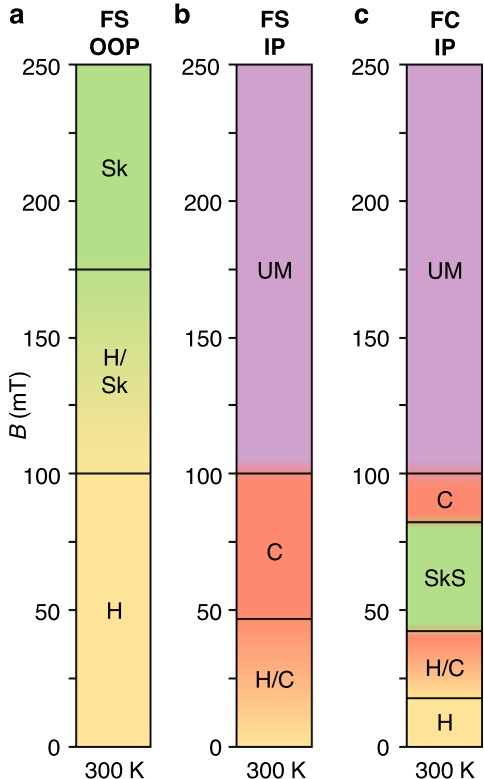

**Fig. 2 Room temperature magnetic phase diagrams. a–c** Diagrams indicating the magnetic state exhibited by the sample as a function of the applied magnetic field when following different measurement protocols: performing a field sweep (FS, − 250 to 250 mT) with an out-of-plane (OOP) magnetic field (**a**), performing a FS with an in-plane (IP) magnetic field (**b**), and finally when field cooling (FC) at a range of IP magnetic fields (**c**). Labels indicate the formation of the helical (H, yellow), conical (C, red), isolated skyrmion (Sk, green), skyrmion string (SkS, green) and uniformly magnetised (UM, purple) spin textures.

of skyrmion states. Plate-like lamellae samples dramatically increase the stability of the skyrmion lattice (SkL) state for OOP fields, extending the equilibrium phase pocket down to lower temperatures far below $T_C$[38]. However, our previous results indicated that for in-plane (IP) fields, the resulting in-plane extended SkS states appeared to be less stable in lamellae than in bulk[11]. The present measurements highlight the wide range of fields over which the SkS remains stable within the nanowire, demonstrating the potential suitability of SkSs for use in confined, device-like geometries.

**Current-induced dynamics**. The possibility to realise current-induced dynamics in the nanowire device was investigated by applying bipolar pulses with a 30 ns duration (15 ns positive current pulse, followed immediately by 15 ns negative current pulse) to single SkS states, as shown in Fig. 4a–j. Bipolar pulses were use to negate any overall current-induced motion of the SkSs along the nanowire. In each case, the current density transmitted through the nanowire during the pulse was estimated to be $5.9 \times 10^{10}$ A/m² (see methods). Fig. 4a–e exhibits an example where an additional SkS was nucleated adjacent to the initial SkS via single pulses. Interestingly, before reaching this state, the system formed a dislocation within the conical background in the vicinity of the original SkS, which in one case was observed to relax back to the conical state during the time between image acquisitions (5 mins). This excited state perhaps forms as an intermediary state on the way to SkS nucleation.

Further measurements showed that SkS states could be reliably generated from an initial conical state by a series of 100 of the same 30 ns bipolar current pulses (full data set shown in Supplementary Fig. S9). In a second example, shown in Fig. 4f–j, we found it was possible to alter the length of the SkS between 3 and 4 conical lengths, $L_C$, with individual pulses of the same current density. This observation therefore represents a current-induced toggle-like switching of the position of an individual Bloch point at the end of the SkS.

To determine the origin of these dynamic effects, we measured the amplitude of the reflected and output (transmitted through the sample) signals as a function of the input pulse amplitude, as shown by the example trace in Fig. 4k. These measurements enable calculation of the resistance of the sample as a function of the input current density (see methods). Together with a calibration of the sample resistance as a function of temperature, this quantifies the change in sample temperature due to the Joule heating effect (more example traces and calibration shown in Supplementary Fig. S10). The resulting plot of temperature change versus current density in Fig. 4l indicates a heating of less than 10 K for the highest amplitude pulses, which is still 40 K below the measured value of $T_C$. Importantly, this is an upper bound estimate of the Joule heating effect, since the calculation assumes that all of the energy was dissipated within the nanowire, whereas in reality heating will also occur within the contact bonds and cables. Therefore, we conclude that a Joule heating effect is likely not a significant factor in the present results, indicating that the observed dynamics are due to some current-induced effect.

The toggle-like motion of the Bloch point indicated that the length of the skyrmion string is in someway quantised to an integer value of the conical length, $L_C$. To investigate this further, we measured the length of the observed SkS states, and plotted their occurrence as a histogram, shown in Fig. 4m. One can see that the majority of the SkS states were close to integer lengths of $L_C$, as indicated by the coloured bands. Similarly, the length of the SkS following each current pulse in Fig. 4f–j is plotted in Fig. 4n, highlighting the toggle-like switching of the Bloch point position by 120 nm. We speculate that this apparent quantisation may occur when it is energetically favourable for the Bloch point to sit at a specific position within the background conical state, perhaps due to the symmetry breaking of magnetocrystalline or shape anisotropy. For SkSs existing within a uniformly magnetised background, we anticipate that such an effect would not be observed. It is likely that such quantisation is possible due to the isotropic 3D DMI interaction found in bulk skyrmion hosts such as the present material. In contrast, in multilayer systems the DMI is strictly 2D due to the interfacial origin of the DMI, and therefore no chiral modulations of the spin texture, such as a conical state or segmented SkS, would be expected along the OOP axis of the system.

**Discussion**

We envisage that this toggle-like switching of individual Bloch points, demonstrated here at room temperature, could have interesting applications in future 3D skyrmion devices utilising SkS states. In Fig. 4o, we present a generalisation of the skyrmion racetrack device concept[39] from 2D to 3D, which could exploit the quantisation of the SkS length within the conical background to store data in the third dimension. A current along the x-axis would translate the SkSs along the wire, as in a typical racetrack design. Individual writing contacts applying current across the z-axis could take advantage of the observed current-induced Bloch point motion to set the length of the SkSs. A second pair of contacts could measure the magnitude of skyrmion Hall effect across the thickness of the wire to read out the length of the stored SkS state.

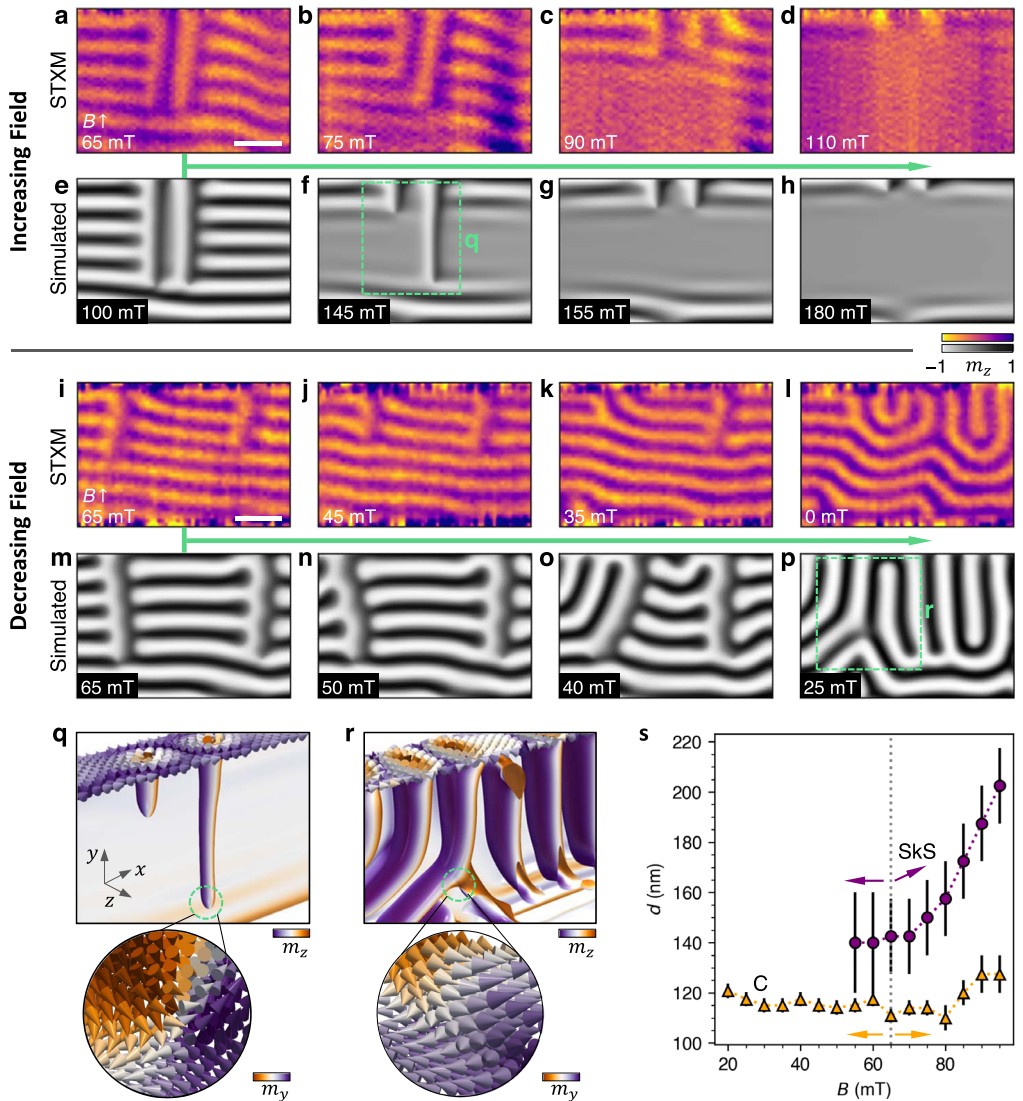

**Fig. 3 Field stability of the skyrmion string spin texture in a nanowire geometry. a–p** X-ray microscopy images (**a-d, i-l**) and simulated images (**e-h, m-p**) of the skyrmion string (SkS) spin texture as a function of increasing (**a-h**), and decreasing (**i-p**) in-plane (IP) applied magnetic field, as indicated by the green arrows. The colourmap indicates the out-of-plane magnetisation component, $m_z$. The scale bar is 250 nm. **q, r** Three dimensional visualisation of the simulations in **f** and **p** respectively, as indicated by the green boxes. The insets show a visualisation of the local spin structure at the end point of the SkS (**q**), and at the formation point of a dislocation (**r**), highlighting the diverging magnetisation indicating the presence of a Bloch point. The colourmap indicates the local magnetisation component, $m_{x,y,z}$. **s,** The experimentally determined spacing $d$ of the SkS (purple circles) and conical (C, orange triangles) states as a function of the applied field. The dotted line and arrows show the initial point and direction of the field sweep. Error bars indicate the error calculated due to the resolution limit, and the averaged number of measured SkS/C spacings.

We note that the current density of $5.9 \times 10^{10}$ A/m² utilised to observe the SkS generation and dynamics here is comparable to current induced motion of 2D skyrmions within lamellae of Co-Zn-Mn alloys[40], and one or two orders of magnitude lower than in examples of skyrmion dynamics within thin-film sputtered systems[5]. The velocity of the Bloch point during the toggling motion, travelling a distance of 120 nm within the 30 ns pulse, can be estimated to be 4 m/s. This is comparable to the 2D skyrmion velocities measured in a lamella of the similar $Co_9Zn_9Mn_2$[40], with maximum velocities of 4 m/s, but is still much lower than the 100 m/s achieved in some multilayer systems[41]. The pulse parameters utilised for the Bloch point toggling correspond to an input energy of 0.12 nJ. In comparison, a typical high-end solid-state drive (SSD) possesses a write speed of 2000 MB/s, and with a power consumption of 8 W, yielding an estimated energy per bit writing of 0.5 nJ. However, the present

system, concerning skyrmion tube length segments of around $100 \times 100 \times 100$ nm in size, is both much slower and much larger compared to a typical element size of less than $10 \times 10 \times 10$ nm in an SSD. The present results feature only bipolar current pulses with a single current density. In the future, we anticipate that increasing the applied voltage amplitude of the current pulse, while simultaneously reducing the pulse length, should increase the velocity of the Bloch point toggling without increasing any heating effects. Moreover, further studies should investigate the effect of monopolar pulses, and also the orientation of the current pulses, for example by applying current pulses locally through the thickness dimension of the nanowire, aiming to achieve finer control of the 3D spin textures.

In summary, the present results feature the formation of long-lived SkS states within a nanowire geometry at room temperature. Furthermore, we demonstrate both single-pulse, current-induced

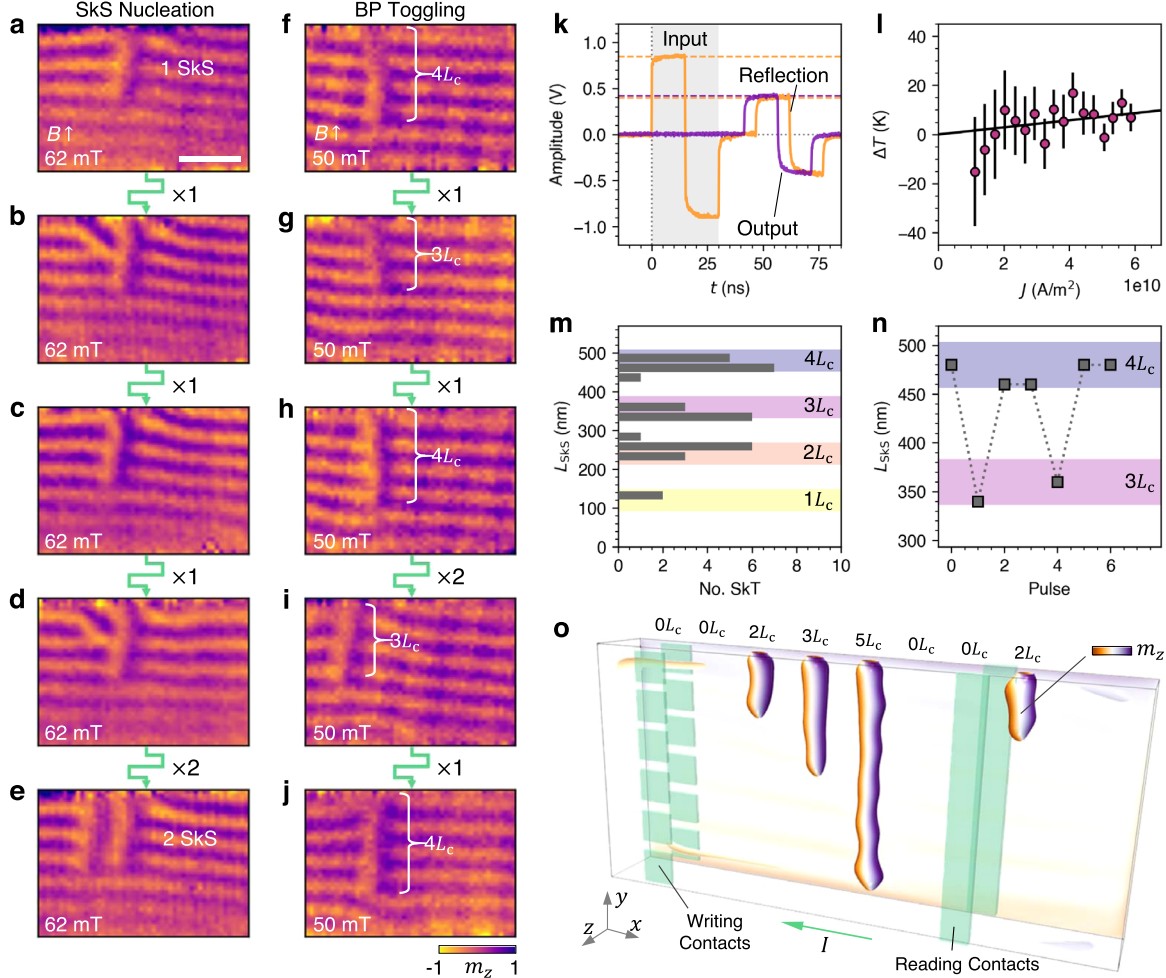

**Fig. 4 Current-induced dynamics of the skyrmion string spin texture at room temperature. a–e** X-ray microscopy images of the nanowire sample, showing the magnetic state following bipolar 30 ns pulses with a current density of $5.9 \times 10^{10}$ A/m², showing the initial single skyrmion string (SkS) state, followed by the formation of conical dislocations and finally a second SkS. The number of pulses between each image is indicated. **f–j** Images of the nanowire sample, showing the magnetic state following bipolar 30 ns pulses of $5.9 \times 10^{10}$ A/m². Successive pulses show alteration of the SkS length between 3 and 4 conical lengths $L_C$, indicating a toggling of the Bloch point (BP) position. The colourmap indicates the out-of-plane magnetisation component, $m_z$. The scale bar is 250 nm. **k** The amplitude of the bipolar pulse trace as a function of time, showing the input (orange), reflected (orange) and output (purple) signals. By measuring the pulse traces for a range of input amplitudes, the resistance of the sample can be calculated, and the temperature change due to Joule heating can be estimated. **l** The estimated change in temperature $\Delta T$ of the sample due to Joule heating as a function of current density $J$, suggesting a $\Delta T < 10$ K. The error bars show the calculated standard error acquired when measuring 10 pulses at each $J$. **m** A histogram showing the measured lengths of SkSs observed within the nanowire sample. Coloured regions indicate integer values of $L_C$, suggesting that the length of the SkS is a quantised integer of $L_C$. **n** The length of the SkS in **f–j** after each consecutive pulse, showing the toggle-like switching of the Bloch point position located at the end of the SkS. **o** An illustration of an example 3D skyrmion racetrack device concept, exploiting the quantised length of the SkS as a data storage element. The colourmap indicates the out-of-plane magnetisation component, $m_z$.

generation of a single SkS, and local manipulation of the SkS texture in the form of toggle-like positional switching of an individual Bloch point located at the end of the SkS. Our analysis confirms that the observed dynamics are most likely current-induced, rather than being caused by Joule heating effects. In the future, we anticipate intensive research into the manipulation of further 3D topological spin textures beyond SkSs, including magnetic hopfions, skyrmion bundles and skyrmion braids, with the view to realise advanced 3D spintronic device architectures.

## Methods

**Sample fabrication and characterisation**. Crystals of $Co_8Zn_9Mn_3$ were grown via the Bridgman method. Stoichiometric amounts of Co powder (Alfa-Aesar, 99.99%), Zn powder (SigmaAldrich, 99.995%) and Mn pieces (Alfa-Aesar, 99.99%) were ground together and transferred to an alumina crucible with a pointed end, and sealed inside an evacuated quartz tube. The tube was then heated to 1060 °C

and allowed to homogenise for 12 hours. It was then slowly cooled at a rate of 1 °C/hr to 700 °C and left to anneal for several days at this temperature before being water quenched. Single crystals of $Co_8Zn_9Mn_3$ were isolated from the as-grown boule and oriented using X-ray Laue back-reflection.

Magnetometry measurements were carried out using an MPMS3 vibrating sample magnetometer, equipped with the AC susceptibility option. A combination of photolithography and magnetron sputtering was performed to pattern ~200 nm thick gold contacts onto a $Si_3N_4$ membrane suitable for transmission measurements. Using a focused ion beam, a thin lamella of material was fabricated using a typical lift-out method. This lamella was milled into a nanowire shape with dimensions of $0.7 \times 0.2 \times 5$ µm. The nanowire was then positioned over the gold contacts on the membrane using a in-situ micromanipulator, and fixed in place with localised platinum deposition.

**Scanning transmission x-ray microscopy**. Scanning transmission x-ray microscopy measurements were carried out at the MAXYMUS endstation at the BESSY II electron storage ring operated by the Helmholtz-Zentrum Berlin für Materialien und Energie. The sample was fixed and bonded to the sample holder, and a

separate resistive wire was attached close to the sample to allow for rapid local heating. With the sample mounted inside the vacuum chamber, the applied magnetic field was achieved by altering the orientation of four permanent magnets. Field-cooling was achieved by applying ~10 s of continuous 1 A current to the resistive wire, which resulted in the temperature of the sample rising above $T_C$, enabling the FC and ZFC procedures to be performed in-situ. Using a Fresnel zone plate and order selecting aperture, the x-ray beam was focused to a ~20 nm spot size. To acquire an image, the sample was rastered through the x-ray focal point using a piezoelectric motor stage, with the transmission measured pixel by pixel. Magnetic contrast was achieved by tuning the x-ray energy to the Co $L_3$ edge at 779 eV, exploiting the resonant absorption effects of x-ray circular dichroism[42]. This results in the absorption of the x-rays changing proportionally to the local out-of-plane magnetisation of the sample, $m_z$. A schematic illustration of the technique and an example XMCD spectrum are presented in Supplementary Fig. S1. All presented STXM images are the result of dividing two images acquired with positive and negative x-ray circular polarisation, eliminating all structural contrast and leaving only the XMCD magnetic contrast. The photon count was measured by an avalanche photodiode. A typical image required 10 min of measurement time for each polarisation. The spatial resolution is estimated to be 20 nm.

Bipolar current pulses were applied to the sample contacts using the two output channels of an Agilent Technologies 81134A pulse generator. The traces of the output and input pulses were recorded using a pickoff T and a Teledyne Lecroy waverunner 8404, such that the signal transmission through the sample could be measured. For the measurements shown in the main text, we applied a 15 ns pulse from each input channel, with the second channel inverted, resulting in a bipolar pulse with a total length of 30 ns and an input pulse amplitude of 0.9 V. For these values, the transmitted output pulse was measured to a 0.41 V measured over the 50 Ω resistance at the oscilloscope. This corresponds to a current density through the nanowire of $5.9 \times 10^{10}$ A/m². For the multipulse excitations, where SkSs were nucleated using a series of 100 pulses, the pulse separation was set to 60 μs – a suitable length to prevent compound heating of the sample from successive current pulses.

A key result of the work is that the observed SkS dynamics are current-induced rather than due to Joule heating effects. To determine this, the change in resistance of the sample, $R_s$, within the x-ray microscope was determined by applying a series of current pulses with input amplitude, $V_{in}$, between 0.1 and 0.9 V, corresponding to current densities between $1.1 \times 10^{10}$ and $5.9 \times 10^{10}$ A/m², and by measuring the amplitude of the reflected and output pulses, $V_{ref}$ and $V_{out}$, respectively. Any increase in sample temperature due to Joule heating would result in an increase in the sample resistance. The resistance can be calculated following the equation[41],

$$R_s = \frac{V_{in} + V_{ref} - V_{out}}{V_{out}} \cdot 50\,\Omega. \quad (1)$$

We then performed a separate resistance versus temperature calibration of the sample on a hot plate, measured by a Keithley 2450 sourcemeter and thermocouple. Utilising this calibration, we converted the change in the sample resistance to a change in the sample temperature as a function of the input pulse current density, $J$. Example pulse traces and the resistance-temperature calibration are shown in Supplementary Fig. S10. The analysis indicates that the upper bound for the temperature change of the nanowire due to the current pulses was less than 10 K. However, this estimate assumes that all of the Joule heating is dissipated within the nanowire itself, while in reality an unknown fraction of the energy will be lost within the contact bonds and cables. Therefore, we concluded that there was likely no significant Joule heating effect within the system for the applied pulse parameters, and certainly not enough to raise the sample to $T_C$.

**Micromagnetic simulations.** Simulations of the SkS states were performed using the micromagnetic finite difference code MuMax3[43] and data was processed using the OOMMFPy library[44]. In the continuum, the cubic chiral-lattice magnet Co₉Zn₉Mn₂, with space group P4₁32, can be described by the energy functional

$$E = \int_V dV \left\{ A \sum_{\alpha=x,y,z} (\nabla m_\alpha)^2 + D\mathbf{m} \cdot (\nabla \times \mathbf{m}) - M_s \mathbf{m} \cdot \mathbf{B} - \frac{M_s}{2} \mathbf{m} \cdot \mathbf{B}_d \right\}, \quad (2)$$

where $\mathbf{m}$ is the normalised magnetisation field, $A$ is the exchange constant, $M_s$ is the saturation magnetisation, $D$ is the DMI constant, $\mathbf{B}$ is the applied field and $\mathbf{B}_d$ is the demagnetising field, including the shape anisotropy of the wire. Thus, the terms within the equation represent the exchange, DMI, Zeeman and demagnetisation energies. The magnetic parameters were adjusted to match the helical lengths observed in the experiments. Specifically, from the dc magnetization data of Co₈Zn₉Mn₃ reported by Bocarsly et al.[45] a saturation magnetization for a cubic lattice of $M_s = 460.55$ kA/m was utilised. Additionally, optimal magnitudes for the exchange and DMI to obtain a helical length of 120 nm were determined by simulating SkS states at different magnetic field strengths, increasing $D$ from an initial value reported by Takagi et al. for Co₉Zn₉Mn₂[46]. As a result, the final exchange and DMI constants utilised were $A = 5.729$ pJ/m and $D = 0.6$ mJ/m², respectively.

The sample was simulated using a rectangular region with dimensions comparable to the experimental system, $5 \times 0.7 \times 0.2\,\mu$m, which was discretised using cubic cells of 5 nm edge length – slightly below the exchange length of the material with a value of 6.56 nm. In order to obtain the SkS states with a Bloch

point at their end, initial states based on a paraboloid function were specified, and the energy was minimised using a combination of the Landau-Lifshitz-Gilbert equation and the steepest descent method. To observe the helical state propagating across the length of the sample as observed in the experiment (rather than across the width), a small anisotropy of 4 kJ/m³ magnitude with a hard axis along the x-axis was included. Simulations using a cubic anisotropy did not show this magnetic configuration. For the simulation of a skyrmion lattice, an initial state based on the magnetization function was used with the helical length of Co₈Zn₉Mn₃[47]. Differences in the applied field values between the simulated and experimental systems can likely be explained due to the temperature effects, where the micromagnetic system is essentially modelled at zero temperature.

## Data availability
The X-ray microscopy, magnetometry and transport data and the relevant analysis scripts utilised to produce the presented figures are available from an online repository[48]. Any further material is available from the corresponding authors upon reasonable request.

## Code availability
Micromagnetic packages MuMax3 and the OOMMFPy Library are available online[43,44]. Simulation data and the relevant analysis scripts utilised to produce the presented simulation figures are available from an online repository[48]. Any further material is available from the corresponding authors upon reasonable request.

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

## Acknowledgements

We thank the Helmholtz-Zentrum für Materialien und Energie for the allocation of synchrotron beamtime. The work was partially funded by the UK Skyrmion Project EPSRC Programme Grant (EP/N032128/1). The authors grateful for the support of A. C. Harrison and U. Eigenthaler for advice on the focused ion beam fabrication.

## Author contributions

M.T.B., P.D.H. and G.S. conceived the project. A.S., D.A.M. and G.B. fabricated the bulk single crystal sample. M.T.B. fabricated the nanowire device structure with focused ion beam. M.T.B., K.L., S.W., F.S. and M.W. performed the x-ray imaging measurements. M.T.B. and KL conducted the current pulse measurements and analysis. D.C.-O. carried out the micromagnetic simulations. M.B. and D.C.-O. wrote the manuscript with input from all authors. All authors discussed the results and gave feedback on the manuscript.

## Funding

## Competing interests

The authors declare no competing interests.
