## [Peer Review File · Nature Communications]

Reviewers' Comments:

Reviewer #1:

Remarks to the Author:

Birch and co-workers report on magnetic real-space imaging using STXM (XMCD) of Skyrmion strings (SkS) within a Co₈Zn₉Mn₃ nanowire. The latter was prepared from a single crystal and fixed to gold contacts using FIB. With the so prepared sample, they performed two main experiments: First, they generated several magnetic states (phase diagrams) by applying in-plane (transverse to the NW axis) and out-of-plane magnetic fields at different protocols (field sweep, field cooling). The magnetic states comprise the helical, Skyrmion lattice, conical and field-polarized phase and some mixtures of them. In a few cases Bloch point domain walls (or topological defects) were stabilized, which are interesting for technological applications such as spintronic devices. In the second part, the dynamics of these magnetic textures are investigated by induction of nanosecond pulses of electrical current. The experimental findings are supported by micromagnetic simulations.

In my opinion, this paper fits perfectly in the scope of Nat. Commun. in terms of its scientific quality and significance. The work reported is very comprehensive and impressive in their experimental realization (e.g. sample preparation, imaging, current-induced dynamics). The results/figures are presented with great care and the paper is very well written as well as the references used are relevant within the context. One concern of mine is that the rather empirical/pragmatic use of micromagnetic simulations (e.g. no amplitude modulation of M, 5 nm cubic cell size, crystalline anisotropy to create helical state (see line 264)) shines enough light to understand in depth the nanostructure of the SkS, Bloch points and their dynamics. I would appreciate if the authors could provide a "more atomistic" view/interpretation, although I am aware that this cannot be explained in a quantitative manner because of its large complexity and size. A second concern is related to current-induced dynamics: Are there reproducible ways to do the SkS nucleation and BP toggling (e.g. using a unipolar (maybe shorter))? Could the authors comment on the expected velocity of these BPs?

Technical questions/comments:

STXM images: Is there a contrast difference between the conical and helical phase visible as the measured component in the helical phase should be stronger? Is it possible to simulate a STXM image (or at least project the OOP component of M) from the micromagnetic simulation? What is the lateral resolution of the STXM images?

Dynamical measurements (lines 214ff): How the power consumption used for BP toggling (~10mW?) compares with state-of-the-art storage elements, when realizing a concept as sketched in Fig. 4o?

Micromagnetic simulations (lines 257ff): Is it necessary to create a larger supercell than the NW dimensions to consider for the demagnetization field (stray field)?

Minor comments:

Line 9: abbreviation of SkS not yet introduced

Line 26 Why the authors introduce Bloch points as topological defects rather than 0D domain walls which sounds more intended and fits in the context of the paper?

Fig. 1c Indication of OOP B-field misleading, because 0mT field.

Line 62: Isn't it the winding vector k a reciprocal length?

Line 65: Is the lattice spacing of the hexagonal structure a lattice vector or a lattice fringe spacing distance as they are geometrically different and the SkL structure could be consistent with superposition of helical modulations by 60°

Lines 152-162: Is it possible that the quantization of the SkS length may be a result of the 3D DMI as present in this helimagnet?

Reviewer #2:

Remarks to the Author:

Birch et al report on "Toggle-like current-induced Bloch point dynamics of 3D skyrmion strings in a room temperature nanowire"

They utilise real-space imaging to investigate the dynamics of a 3D SkS within a nanowire of $\text{Co}_8\text{Zn}_9\text{Mn}_3$, which hosts skyrmions at room temperature. They use current pulses to demonstrate current-induced nucleation of a single SkS, and a toggle-like positional switching of an individual Bloch point at the end of a SkS.

Control of skyrmion string in a "nanowire" and the possibility to create skyrmions strings with current pulses are definitely very interesting results and in my opinion are suitable for publications in Nat Comm.

However, I have few concern about the procedure that is used to infer the presence of the SkS after the current pulses are applied.

Namely, the author claims that the periodicity of the SkS is larger than the one of the Conical/helical phase.

For example, can they exclude that larger period observed is not due to inhomogeneity in the chemical composition of the sample and/or gradient in the thickness of the sample?

Also if I look at Extended Data Figure 2, it looks as the sample is not showing an homogenous response as in Fig 1.

For example in one case one gets a SkL while in the other case isolated skyrmions.

Why this is the case? Is it a matter of following a specific protocol with temperature and magnetic field?

If this is the case, I would suggest to expand on this in the paper text.

I have also the following comments/questions for the authors:

Why the author have chosen STXM over electron microscopy ?

line9: please introduce acronyms

line10: I find the word "nanoscale" not ideal in this context, perhaps the authors could use "nA" ?

line36-40: I find this 2D vs 3D confusing as the skyrmions in real materials are inherently 3D object.

Probably better to reformulate the text to convey that they use SK with a length > than few nm or that

the hosting material is "2D" like ...

Fig 1c-f: Field in c,d is the same in exp/sim. Why it is not the case for e,f? In the SI it is mentioned,

that temperature effect are not included in the simulation and could explain the difference.

However, I wonder if one can go a step further, for example discussing while a substantial difference is

observed only for the skyrmion lattice (SkL) phase.

Fig1g-h: I find it difficult to understand the color code. Colorbar for m_x , m_y and m_z components use the same color...

Fig 1f: if I look at the STXM experimental data the SkS on the left matches very well with the simulation, while for the string on the right it looks like there is a region where the SkS is not connected to the region hosting the conical state.

Any idea why this is the case?

Fig 1f: The presence of the Bloch point can be inferred by the STXM data alone?

line62: "winding vector k ", this notation is a little bit confusing, as " k " is commonly use for reciprocal space... maybe "period" would be preferable?

line 64: Please define acronyms in the body part of the paper

line 73: "highlighting the Bloch-type chiral structure of the skyrmions" : not very clear from the image...

Fig 2: Perhaps space could be use more efficiently in this figure... (spell out phase with larger columns or make rows rather than columns...)

line 128: "relax back to the conical state with time". Would it be possible to write "numbers" (hours/days...)?

line 130: "reproducibly", also here it will be nice to have numbers.

line 264: "a small anisotropy of 4 kJ/m³ magnitude with a hard axis along the x-axis was included". Any idea on what would be the reason for the need to introduce this extra term?

Extended Data Figures 2/3 Guess this was done at 300K as specified in Fig 2, but it will be nice to have it in the caption as well.

Extended Data Figure 8: what is the protocol followed before

Extended Data Figure 9: typo "Concial"

Reviewer #1:

“Birch and co-workers report on magnetic real-space imaging using STXM (XMCD) of Skyrmion strings (SkS) within a Co₈Zn₉Mn₃ nanowire. The latter was prepared from a single crystal and fixed to gold contacts using FIB. With the so prepared sample, they performed two main experiments: First, they generated several magnetic states (phase diagrams) by applying in-plane (transverse to the NW axis) and out-of-plane magnetic fields at different protocols (field sweep, field cooling). The magnetic states comprise the helical, Skyrmion lattice, conical and field-polarized phase and some mixtures of them. In a few cases Bloch point domain walls (or topological defects) were stabilized, which are interesting for technological applications such as spintronic devices. In the second part, the dynamics of these magnetic textures are investigated by induction of nanosecond pulses of electrical current. The experimental findings are supported by micromagnetic simulations.

In my opinion, this paper fits perfectly in the scope of Nat. Commun. in terms of its scientific quality and significance. The work reported is very comprehensive and impressive in their experimental realization (e.g., sample preparation, imaging, current-induced dynamics). The results/figures are presented with great care and the paper is very well written as well as the references used are relevant within the context.”

1) *“One concern of mine is that the rather empirical/pragmatic use of micromagnetic simulations (e.g., no amplitude modulation of M , 5 nm cubic cell size, crystalline anisotropy to create helical state (see line 264)) shines enough light to understand in depth the nanostructure of the SkS, Bloch points and their dynamics. I would appreciate if the authors could provide a “more atomistic” view/interpretation, although I am aware that this cannot be explained in a quantitative manner because of its large complexity and size.”*

The referee is correct that the primary use of the micromagnetic simulations in our work is to verify the identity of the skyrmion tubes within the experimental images. Since we are not attempting to fully reproduce the magnetic phase diagram, we have not included thermal effects. However, the size of the micromagnetic cells is justified by being smaller than the exchange length of the material, and also much smaller than the characteristic helical/conical winding length of 120 nm. The referee is right that performing atomistic investigations on a system of comparable size to the experimental sample would be challenging, as well as extremely computationally demanding.

We actually have published a previous paper performing atomistic simulations on the dynamics of skyrmion annihilation to either the conical or helical states (reference 17), using the nudged elastic band method to investigate the energy barrier of the Bloch point unwinding mechanisms. Perhaps this article would satisfy the referees request for a deeper understanding of the Bloch point-mediated skyrmion dynamics. In this previous work, we modelled the behaviour using energy parameters with dimensionless units, looked at the annihilation of only a single skyrmion string in a 30x30x30 cell box, and only captured three conical/helical winding lengths. Due to the complexity of the simulations, it was necessary to downscale the system and look only at general effects rather than a one-to-one comparison to the experimental system. The experimental results of the present manuscript are most similar to the atomistic simulations of the skyrmion string annihilation into the conical or uniformly magnetised states.

It is also true that in the present simulations, the energy of the Bloch points may not be properly represented due to its small size, as in [Thiaville et al. *Phys. Rev. B* **67**, 094410 (2003)]. A thorough investigation of these cell-size considerations would require significant additional work beyond the scope of the present manuscript. Nevertheless, it would certainly be interesting to simulate and observe the effects of the Bloch point on the end of the SkS due to thermal noise and the applied current. One way to tackle this might be to use a combination

of micromagnetics with atomic discretisation around the Bloch point itself, similarly to a previous work [Andreas et al. *J. Magn. Magn. Mater.* **362**, 7-13 (2014)].

2) “A second concern is related to current-induced dynamics: Are there reproducible ways to do the SkS nucleation and BP toggling (e.g., using a unipolar (maybe shorter))?”

The referee has identified one of the shortcomings of the paper: the same kind of single bipolar pulse resulted in both SkS nucleation and Bloch point toggling. If we had more beamtime, we would like to investigate the possibility to tune the pulse length and experiment with monopolar pulses to see if finer control over the dynamics can be realised. Unfortunately for now this must wait until we acquire more time for this project at the synchrotron. We have added an acknowledgement of this limitation, and details of suggested future work in the discussions section (lines 196-203), pointing out that further investigations are required to achieve a finer control of the spin textures.

3) “Could the authors comment on the expected velocity of these BPs?”

The referee is right that this would be interesting to include in the manuscript. If we take the change in Bloch point position as 120 nm, and the pulse time of 15 ns, then we can estimate that the Bloch point moves with at least 8 m/s velocity. It would be interesting to see if a higher amplitude, but shorter duration, could improve upon this number. This compares to 2D skyrmion velocities in nanostructures of the similar $\text{Co}_9\text{Zn}_9\text{Mn}_2$ bulk DMI material of 4 m/s [Peng et al. *Nat. Commun.* **12**, 6797 (2021)], or skyrmion velocities in sputtered thin films of ~100 m/s [Litzius et al. *Nat. Electron.* **3**, 30-36 (2020)]. We have included details of this discussion in a new paragraph at the end of the manuscript (lines 188-192), before the final summary paragraph.

4) “Is there a contrast difference between the conical and helical phase visible as the measured component in the helical phase should be stronger?”

There is indeed a contrast difference between the helical and conical phase. The amplitude of the m_z component of the conical phase observed with the XMCD signal reduces as a function of the applied in-plane field, as the spins within the conical state tilt towards the applied field direction. To demonstrate this, we have analysed the in-plane field sweep data in Extended Data Fig. 4 to extract the XMCD contrast amplitude exhibited by the helical stripes at 20 mT, and the conical stripes between 40 and 80 mT. The resulting plot of the average percentage out-of-plane XMCD contrast, proportional to m_z , is shown below. Due to the slight thickness gradient of the sample (~20 nm difference in top and bottom edges), the conical stripes at the bottom of the wire lose contrast at lower fields in comparison to the top of the wire, but the overall average contrast of each stripe shows a clear decrease with increasing in-plane field. We have added this plot to Extended Data Fig. 4.

5) “Is it possible to simulate a STXM image (or at least project the OOP component of M) from the micromagnetic simulation?”

The images of the micromagnetic simulations presented in the main text do indeed show the average projected magnetisation component along the observation axis, m_z , as the referee

suggests, but we had not done a good job explaining this. We have edited the main text to make it clear that this is the case (line 75). Moreover, this will be directly explained and demonstrated in the notebook used to analyse the simulations accessible in the online repository (now available online).

6) *“What is the lateral resolution of the STXM images?”*

The resolution of the STXM images is determined primarily by the Fresnel zone plate utilised at the beamline, but also the energy of the x-rays and the width of the exit slits on the beamline. With the settings used (zone plate with outer-most zone 18 nm, x-ray energy of ~780 eV and nominally 25 μm exit slits), we estimate that the resolution is approximately 20 nm. Improving the resolution beyond this is possible (to a maximum of 5-6 nm), but results in significantly lower focused beam intensity. We have added the spatial resolution information to the methods section (line 246).

7) *“Dynamical measurements (lines 214ff): How the power consumption used for BP toggling (~10mW?) compares with state-of-the art storage elements, when realizing a concept as sketched in Fig. 4o?”*

For the given pulse parameters where Bloch point motion was observed, with transmitted pulse amplitude of 0.9 V, length 15 ns and with a sample resistance of ~100 Ω , we calculated the energy of the pulse was ~0.12 nJ, giving a power consumption of 8 mW – very close to the referee’s estimation of 10 mW. For a high-end commercial SSD, with a writing speed of 2000 megabytes/s and a power consumption of 8 W, an estimation of the energy for each bit would be 0.5 nJ. However, of course the quantized segments of the skyrmion tube have dimensions of about 100x100x100 nm, in comparison to the ~10x10nm dimensions of the bits within SSDs, so we are still some ways away from the state of the art. Nevertheless, we have included this discussion in a new paragraph at the end of the manuscript, before the final summary paragraph (lines 192-196).

8) *“Micromagnetic simulations (lines257ff): Is it necessary to create a larger supercell than the NW dimensions to consider for the demagnetization field (stray field)?”*

The demagnetising field in our case is computed by a standard method in finite difference micromagnetics, where the demagnetising field of every cuboid is computed analytically, resulting in a mathematical convolution expression that can be solved numerically, and very efficiently, using a fast Fourier transform. This procedure is already implemented directly within the MuMax3 code. Since we are using a finite, confined sample (rather than periodic boundaries), the demagnetising field should be calculated correctly without needing a larger supercell for the system. A different approach is required when using periodic boundary conditions, when it is necessary to consider many “neighbouring cells”, and therefore consider some form of supercell structure, to account for a potentially infinite sample, but in the present case this should not be necessary.

9) *“Line 9: abbreviation of SkS not yet introduced”*

We have fixed this issue, introducing the abbreviation in the abstract of the paper and the main body of text.

10) *“Line 26 Why the authors introduce Bloch points as topological defects rather than 0D domain walls which sounds more intended and fits in the context of the paper?”*

Previous papers have described Bloch points as topological defects – the mediators of skyrmion crystal annihilation [Kagawa et al, *Nat. Commun.* **8**, 1332 (2017); Yu et al, *Nano Lett.* **20**, 7313-7320 (2020)]. In this way, analogies have been drawn to the description of other topological defect-mediated phase transitions, such as the proliferation and propagation of dislocations or disclinations in the melting of crystalline solids (we take inspiration from the

very nice early review of topological defects in condensed matter by Mermin [Mermin, *Rev. Mod. Phys.* **51**, 591 (1979)]. However, the referee is right that a 0D domain wall is also a fitting description of a Bloch point, and perhaps more relevant in the context of magnetic systems. Therefore, we have altered the main text to include this description (line 27).

11) "*Fig. 1c Indication of OOP B-field misleading, because 0mT field.*"

We have removed the OOP field indicator for Fig. 1c.

12) "*Line 62: Isn't it the winding vector k a reciprocal length?*"

The referee is correct here. To clear up this, we now refer to the winding vector k , and the real-space winding length of 120~nm separately (line 65-66).

13) "*Line 65: Is the lattice spacing of the hexagonal structure a lattice vector or a lattice fringe spacing distance as they are geometrically different and the SkL structure could be consistent with superposition of helical modulations by 60°* "

The referee is right that this could cause some confusion. The superposition of three helical modules at angles of 60° with winding length 120 nm forms a triangular lattice of skyrmions, with a skyrmion-to-skyrmion separation distance of ~148 nm, but with a lattice spacing 120 nm. We have updated the main text to specify that the 148 nm is the skyrmion separation distance (line 68).

14) "*Lines 152-162: Is it possible that the quantization of the SkS length may be a result of the 3D DMI as present in this helimagnet?*"

This is certainly an interesting possibility, and we suppose it could be similar to the 3D hedgehog skyrmion lattice proposed in MnGe samples [Tanigaki et al, *Nano. Lett.* **15**, 5438-5442 (2015)]. The referee's point also raises the question as to whether objects such as the chiral bobber would be stable in a suitably thin sputtered multilayer system, where the DMI is strictly 2D due to the interfacial origin of the DMI. Certainly, it is not possible to stabilise an out-of-plane modulation such as the conical state within such thin film systems. Therefore, it's possible that the anisotropic 3D DMI is required for the chiral bobbers, and in turn also the quantisation of the SkS. It's something we had not considered before, and therefore we have added some of these ideas to the main text (lines 171-175).

We would like to thank the referee for their appraisal of our work, as well their many insightful comments and suggestions. In particular, we feel the new discussion paragraph comparing the energies and velocities of the Bloch point dynamics to other systems is an excellent addition, placing our results better in the context of skyrmion-based device concepts, and was directly inspired by their comments. We hope the referee is satisfied with the other explanations and clarifications, and finds the new version manuscript greatly improved.

Reviewer #2:

“Birch et al report on “Toggle-like current-induced Bloch point dynamics of 3D skyrmion strings in a room temperature nanowire”. They utilise real-space imaging to investigate the dynamics of a 3D SkS within a nanowire of Co₈Zn₉Mn₃, which hosts skyrmions at room temperature. They use current pulses to demonstrate current-induced nucleation of a single SkS, and a toggle-like positional switching of an individual Bloch point at the end of a SkS. Control of skyrmion string in a “nanowire” and the possibility to create skyrmions strings with current pulses are definitely very interesting results and in my opinion are suitable for publications in Nat Comm.”

1) “However, I have few concern about the procedure that is used to infer the presence of the SkS after the current pulses are applied. Namely, the author claims that the periodicity of the SkS is larger than the one of the Conical/helical phase. For example, can they exclude that larger period observed in not due to inhomogeneity in the chemical composition of the sample and/or gradient in the thickness of the sample?”

The referee is right to question our method of determining the presence of the SkS state. We believe we can exclude the possibility that sample inhomogeneities could result in the observed changes in the winding length of the spin texture. The figure below shows the STXM images acquired after 100x pulses at 0 and 60 mT, where purple and orange lines indicate the positions of the line scans plotted in panel c. As can be seen from the images, the two line scans are taken in the same position within the wire, and are the same length of ~850 nm. Examination of the line profile shows 5 peaks (white lines in the image) for the spin texture in a, but 6 peaks for the spin texture in b. Thus, despite both exhibiting vertical contrast, and being located in the same position in the sample, the spin textures have different characteristic winding lengths of 145 nm and 120 nm respectively. Since the spin textures are observed in the same location, we believe this demonstrates that there cannot be a sample inhomogeneity leading to the change in the observed feature sizes.

It is known that for isolated skyrmions within a conical or uniformly magnetised background, the separation distance of neighbouring skyrmions can vary depending on the field [Du et al, *Phys. Rev. Lett.* **120**, 197203 (2018)] (reference now included in the main text, reference 36). We directly demonstrate this behaviour by extracting the separation distance of the skyrmion strings and the conical/helical winding length as a function of field in the micromagnetic simulations. The resulting data, shown below, reveals a comparable behaviour to the experimental SkS separation distance as a function of field seen in Fig. 3s. We have included

this plot in an updated Extended Data Fig. 9. The larger separation of the skyrmion strings compared to the conical winding length seems to be a reliable method to distinguish skyrmion states from the helical and conical states.

2) “Also if I look at Extended Data Figure 2, it looks as the sample is not showing an homogenous response as in Fig 1. For example in one case one gets a SkL while in the other case isolated skyrmions. Why this is the case? Is it a matter of following a specific protocol with temperature and magnetic field? If this is the case, I would suggest to expand on this in the paper text.”

We admit that, as initially presented, the fact that no skyrmion lattice forms during the out-of-plane field sweep process in Extended Data Fig. 2, while we present an image of a full skyrmion lattice in Fig. 1, was confusing. The reason for these different observations is because the nice skyrmion lattice state in Fig. 1 was prepared by field-cooling the nanowire from above T_C under an out-of-plane applied magnetic field of 80 mT. The ordered hexagonal skyrmion lattice forms in the high temperature equilibrium skyrmion pocket during the cooling process, and is preserved to room temperature. In comparison, the field-sweep procedure at room temperature results in only a few isolated skyrmions forming, as shown in Extended Data Fig. 2. We did not do a good job of explaining this in the first draft, so we have updated the main text to mention this directly (lines 89-91).

3) “Why the author have chosen STXM over electron microscopy?”

There are a couple of reasons we chose to use STXM over Lorentz Transmission Electron Microscopy (LTEM). The first is simply a practical reason: we have reasonable access to the STXM at BESSY, while lacking direct access to an electron microscope capable of LTEM measurements ourselves. Secondly, it can be quite challenging to apply an in-plane applied magnetic field during LTEM measurements, mainly due to the resulting deflection of the probe electrons due to the Lorentz force, and also because applying a uniform in-plane field requires additional equipment to be prepared for the LTEM holder (the objective lens is typically used to apply out-of-plane magnetic fields during LTEM imaging). In comparison, the x-ray Microscope at BESSY is equipped with an array of four permanent magnets which allow for the field to be quickly changed between out-of-plane and in-plane field during measurements, and the x-rays are not influenced by the orientation of the applied field. Despite this, other groups have observed SkS using LTEM measurements (references 12 and 13). Despite the additional challenges, the spatial resolution of LTEM is certainly better than the STXM, so it would be interesting to try LTEM measurements on a similar nanowire sample in the future.

4) “line9: please introduce acronyms”

We have updated the abstract and main text to introduce the SkS acronym correctly.

5) “line10: I find the word “nanoscale” not ideal in this context, perhaps the authors could use “nA”?”

The referee is right that the word nanoscale seems to imply a length, rather than the time duration we were trying to imply. We decided the easiest solution was to omit the word altogether, so the abstract now reads “Utilising single current pulses, we demonstrate current-induced nucleation of a single SkS...”

6) “line36-40: I find this 2D vs 3D confusing as the skyrmions in real materials are inherently 3D object. Probably better to reformulate the text to convey that they use SK with a length > than few nm or that the hosting material is “2D” like ...”

Referee is right to draw this distinction, as skyrmions in multilayer systems still have a vertical size. We have reworded this section as the referee suggests, by referring to the skyrmions in thin systems as “2D-like” and included a rough definition that the vertical size of such 2D-like skyrmions should be less than their lateral size (lines 39-40).

7) “Fig 1c-f: Field in c,d is the same in exp/sim. Why it is not the case for e,f? In the SI it is mentioned, that temperature effect are not included in the simulation and could explain the difference. However, I wonder if one can go a step further, for example discussing while a substantial difference is observed only for the skyrmion lattice (SkL) phase.”

The simulation of the SkL in Fig. 1e was created by relaxing an initialised skyrmion lattice state at a range of applied fields (results shown below). It is challenging to realise such a lattice state via a random configuration and field manipulation of the simulated system, since the formation of the SkL requires high temperatures which can be difficult to model in micromagnetic simulations. From here, we simply chose the state which appeared to be most comparable to the experiment – the 120 mT image. Just as the SkL in the simulation can be realised at a range of applied fields, in the experiment we can also expect that a full SkL could be realised when field-cooling at range of out-of-plane applied fields, so the specific field should not matter too much.

The referee is correct that there is a discrepancy between the field at which the sample saturates in the experiment and the same field point in the simulated system. It generally required a higher applied field to achieve saturation in the simulated system, and we expect this is largely down to the simulation being effectively performed at 0 K, as the referee suggests. Perhaps the reason for the greater difference in the SkL state may be that there is a larger discrepancy in the saturation field for the OOP compared to the IP field orientation.

8) *"Fig1g-h: I find it difficult to understand the color code. Colorbar for m_x , m_y and m_z components use the same color..."*

We tried to alter this by changing the colour maps used for each component in the visualisation program, Paraview. However, it appeared to us that it wasn't possible to change the colour map for individual elements of the visualisation, due to the limitations in the program (or our own ability to manipulate it). To aid in the interpretation of the spin directions, we have added arrows to indicate the spin directions of the skyrmion contour sections on Fig. 1g and h. We hope this goes some way to help visualise the 3D spin texture.

9) *"Fig 1f: if I look at the STXM experimental data the SkS on the left matches very well with the simulation, while for the string on the right it looks like there is a region where the SkS is not connected to the region hosting the conical state. Any idea why this is the case?"*

The referee is right to point this out. We went back to the raw images of each XMCD polarisation to investigate this matter further, plotted below with a background subtraction to remove as much of the structural contrast as possible. We have also reversed the contrast of the C- image so that the contrast of each feature matches. In both the left and right circularly polarised (C+ and C-) images, it seems like the conical state extends right up to the SkS as in the simulations. However, in the subtracted XMCD image, you can see the absence of conical contrast in the vicinity of the end of the SkS as the referee pointed out. If we look at the conical stripes between the two SkS closely, it looks like the conical stripes have rearranged between the acquisition of the two polarisations. This results in the smearing of the conical state contrast around the SkS in the difference (XMCD) image.

We have made a note of this issue in the caption of Figure 1.

10) *"Fig 1f: The presence of the Bloch point can be inferred by the STXM data alone?"*

The referee is correct that the presence of a Bloch point is not directly evident from the STXM image in 1f. This is one of our primary motivations for including micromagnetic simulations of states comparable to those observed in experiment – the understanding of the structure in the simulations allows us to infer details of the experimental images. Since in reality the size of the Bloch point is likely to be on the order of a few spins, resolving the structure directly with imaging will be challenging without a dramatic improvement of the spatial resolution. We have added a couple of sentences to the main text to discuss this issue (lines 80-83).

11) *"line62: "winding vector k ", this notation is a little bit confusing, as " k " is commonly use for reciprocal space... maybe "period" would be preferable?"*

The referee is absolutely correct here, and as written this is confusing. To clear up this, we now refer to the winding vector k , and the real-space winding length of 120 nm separately (lines 65-66)

12) *“line 64: Please define acronyms in the body part of the paper”*

We have added the definition of the XMCD acronym (line 63), and for OOP and IP acronyms (line 57) to the main text.

13) *“line 73: “highlighting the Bloch-type chiral structure of the skyrmions” : not very clear from the image...”*

The referee is right that the original visualisation of the skyrmion using cones for the magnetisation vectors was not very clear. We have replaced the inset with a new image using arrows which we hope better displays the Bloch-type spin texture of the simulated skyrmion.

14) *“Fig 2: Perhaps space could be use more efficiently in this figure... (spell out phase with larger columns or make rows rather than columns...)”*

Our intention was for this figure to be a single column image, and therefore it wouldn't take up so much space as in a double column figure. Hopefully this can be specified if the manuscript is accepted for publication.

15) *“line 128: “relax back to the conical state with time”. Would it be possible to write “numbers” (hours/days...)?”*

The relaxation was between two images of opposite polarisation, so within 5 minutes of the dislocation being formed. We have updated the main text to add this detail (line 141).

16) *“line 130: “reproducibly”, also here it will be nice to have numbers.”*

The data presented in Extended Fig. 8 shows the vast majority of our attempts to nucleate SkSs using the 100x pulsed method. Due to the constrained nature of beamtime experiment, we were not able to take a large sample size to determine just how likely SkS nucleation is. For this reason, as the referee indicates, we decided that perhaps the word “reproducibly” was too strong, so we decided to change some instances to “reliably” (line 143), and in other cases just deleted the word altogether. All attempts at SkS nucleation by 100x pulses within 40-60 mT resulted in SkS formation, but the number of instances of this measurement is perhaps around 5 only. We hope the referee is satisfied by this change.

17) *“line 264: “a small anisotropy of 4 kJ/m³ magnitude with a hard axis along the x-axis was included”. Any idea on what would be the reason for the need to introduce this extra term?”*

We agree that this is quite a curious requirement. Typically, in B20 skyrmion materials, the orientation of the helices is determined by the interplay of the cubic anisotropy with the exchange and DMI energies, and in some cases, such as the tilted conical state found in Cu₂OSeO₃ this can result in quite complex behaviour [Halder et al. *Phys. Rev. B* **98**, 144429 (2018)]. Due to the small magnetocrystalline anisotropies found in B20 materials, in cases with confined sample geometries the helical orientation is typically dominated by the shape anisotropy, or the demagnetising field, of the sample. In the present case of a nanowire, this results in the helices lying along the length of the nanowire, as shown by the state after the ZFC process shown in Extended Data Fig. 4a.

However, in our initial simulations without the added anisotropy term, we found the system preferred to form with the helical vector along the shortest axis – across the thickness of the system. We attach example simulated images of the system with -1 and -4 kJ/m³ below. Here, the system was initialised with two SkSs together, and then minimised at different decreasing field values.

One can see that for the smaller value of uniaxial anisotropy, at low fields the sample forms a helical state winding in the out-of-plane direction (grey contrast). The addition of a slightly larger hard axis anisotropy more closely matches the experimental behaviour. We also tried utilising cubic anisotropy in the simulations and were unable to reproduce the experimental behaviour. However, once again, since our primary aim with the simulations was to assist with interpretation of the STXM images, we chose the pragmatic approach of employing the uniaxial anisotropy to reproduce the experimental data as closely as possible.

18) *“Extended Data Figures 2/3 Guess this was done at 300K as specified in Fig 2, but it will be nice to have it in the caption as well.”*

The referee is correct, we have updated the relevant captions to make this clear across all extended data figures.

19) *“Extended Data Figure 8: what is the protocol followed before”*

We have updated the figure caption to explain in more detail how the initial state (before pulse) was prepared.

20) *“Extended Data Figure 9: typo “Concial”*

This typo has now been corrected.

We would like to thank the referee for their careful and detailed reading of our manuscript, and for pointing out many points of possible confusion. We found that their main concern of our work was the method by which we determined the presence of the SkS spin structure. We hope that our explanations and additions to the article can resolve these concerns, and that they find the newest version of the manuscript greatly improved, and suitable for publication.

Reviewers' Comments:

Reviewer #1:

Remarks to the Author:

The authors have replied very thoroughly and detailed to all my concerns, and have modified/improved the manuscript accordingly. I appreciate the new discussion paragraph comparing the energies and velocities of the Bloch point dynamics to other systems. I am glad that I could contribute to this excellent paper in this way.

Reviewer #2:

Remarks to the Author:

I'm happy with the response of the authors and the changes that they have implemented to the manuscript.

Reviewer #1

"The authors have replied very thoroughly and detailed to all my concerns, and have modified/improved the manuscript accordingly. I appreciate the new discussion paragraph comparing the energies and velocities of the Bloch point dynamics to other systems. I am glad that I could contribute to this excellent paper in this way."

Reviewer #2

"I'm happy with the response of the authors and the changes that they have implemented to the manuscript."

We are pleased that both reviewers are satisfied with the changes we have made to the manuscript following their feedback and comments, and thank them once again for their contributions to the improvement of the article.